# The Clinical Findings, Pathogenic Variants, and Gene Therapy Qualifications Found in a Leber Congenital Amaurosis Phenotypic Spectrum Patient Cohort

**DOI:** 10.3390/ijms25021253

**Published:** 2024-01-19

**Authors:** Richard Sather, Jacie Ihinger, Michael Simmons, Glenn P. Lobo, Sandra R. Montezuma

**Affiliations:** Department of Ophthalmology and Visual Neurosciences, University of Minnesota Medical School, Minneapolis, MN 55455, USA; richardsather8@gmail.com (R.S.III); jacqueline.ihinger@fairview.org (J.I.); lobo0023@umn.edu (G.P.L.)

**Keywords:** Leber congenital amaurosis, early-onset severe retinal dystrophy, inherited retinal disease, genetic testing, gene therapy, ocular coherence tomography, fundus autofluorescence

## Abstract

This retrospective study examines the clinical characteristics and underlying genetic variants that exist in a Leber congenital amaurosis (LCA) patient cohort evaluated at the inherited retinal disease (IRD) clinic at the University of Minnesota (UMN)/M Health System. Our LCA cohort consisted of 33 non-syndromic patients and one patient with Joubert syndrome. We report their relevant history, clinical findings, and genetic testing results. We monitored disease presentation utilizing ocular coherence tomography (OCT) and fundus autofluorescence (FAF). Electroretinogram testing (ERG) was performed in patients when clinically indicated. Next-generation sequencing (NGS) and genetic counseling was offered to all evaluated patients. Advanced photoreceptor loss was noted in 85.7% of the subjects. All patients who underwent FAF had findings of either a ring of macular hypo/hyper AF or peripheral hypo-AF. All patients had abnormal ERG findings. A diagnostic genetic test result was identified in 74.2% of the patients via NGS single-gene testing or panel testing. Two patients in our cohort qualified for Luxturna^®^ and both received treatment at the time of this study. These data will help IRD specialists to understand the genetic variants and clinical presentations that characterize our patient population in the Midwest region of the United States.

## 1. Introduction

Leber’s congenital amaurosis (LCA) is the earliest and most severe form of the inherited retinal dystrophies (IRDs) [1]. The approximate birth prevalence of this dystrophy is rare at two to three cases per 100,000 births [2]. LCA should be suspected in the pediatric population when vision loss is noted in infancy. An ophthalmological examination may reveal blindness or impaired vision that appears in the first year of life, a sluggish pupillary response to light, nystagmus, oculo-digital signs (eye rubbing, poking, prodding, etc.), an absent or severe reduction in scotopic and photopic electroretinograms (ERGs), abnormal visual-evoked potentials (VEPs), and variable fundus appearance. Overall, LCA represents nearly 5% of IRDs, although it may be more prevalent in regions with higher rates of consanguinity [3]. The differential diagnosis for LCA also includes achromatopsia, S-cone monochromatism, congenital stationary night blindness, and albinism.

Histopathological studies indicate that LCA is degenerative in nature rather than ageneic, with the principal aberrations involving the susceptible outer retina and photoreceptor layers [4]. In most cases, the formation of 11-cis-retinal in the phototransduction pathway is impaired [5]. As a result, downstream enzyme reactions in the phototransduction cascade are impaired, preventing neuronal signals from reaching the visual cortex. There is, perhaps, a link to improper vitamin A metabolism and photoreceptor degeneration [6].

LCA is primarily inherited in an autosomal-recessive manner. Various distinct phenotypes have been described (LCA1-19) as being associated with genetic variants in at least 29 different genes. The most frequent genetic causes of LCA are attributed to the *CEP290* (15%), *GUCY2D* (12%), *CRB1* (10%), and *RPE65* (8%) genes and account for 70–80% of known cases [5].

It is important to recognize other historical terms for the LCA phenotype. Theodore Leber first described LCA in 1869 as a recessively inherited group of severe, early-onset rod–cone dystrophies that present in infancy [7]. In the early 20th century, Leber then observed a milder form of LCA, which has been called by multiple names in the literature, including early-onset severe retinal dystrophy (EOSRD), severe early childhood-onset retinal dystrophy (SECORD) [8], and early-onset RP. One difference between LCA and the other forms mentioned is the timing of the initial presentation. LCA is often present in the first few months of life, while EOSRD/SECORD present sometime before the age of five. As previously mentioned, LCA usually presents with nystagmus, poor pupillary response, and severely reduced ERG response, while EOSRD/SECORD have better visual functioning and small ERG signals. There is overlap in the genes that cause LCA and EOSRD, but some patterns exist: the *GUCY2D*, *NMNAT1*, *CEP290*, and *AIPL1* genes are more likely to cause LCA, whereas *RPE65*, *LRAT*, and *RDH12* are more commonly associated with the EOSRD phenotype [5]. Still, these are generalizations; the variable nomenclature for LCA reflects the broad phenotypic variability that is seen among patients with LCA. Overall, a clear definition of LCA based on clinical and genetic heterogeneity is lacking [9,10].

*GUCY2D* was the first documented gene linked to LCAA. Variants in this gene account for approximately 10–20% of cases of LCA [11]. GUCY2D encodes the enzyme retinal guanylate cyclase-1 (RetGC1) located in the outer photoreceptor layer and functions to recover photoreceptors following phototransduction. Pathogenic variants in this gene mimic chronic light exposure [11] with foveal cone loss [12] that display markedly reduced visual acuity and color perception, and overall profound nyctalopia [13,14].

Fifteen to twenty percent of LCA cases are caused by biallelic pathogenic variants in the *CEP290* gene [15], which encodes a protein localized to the centromeres and cilia of photoreceptors. Patients with these variants have a distinct phenotype characterized by a normal-appearing fundus and variably affected visual acuity in the first decade of life [16].

Variants in the *CRB* gene account for approximately 10% of LCA cases with a broad range of associated phenotypes, including LCA, EOSRD, and early-onset RP [17]. *CRB* encodes a protein that forms a major component of the outer limiting membrane. It has possible involvement in retinal development [18]. The rate of disease progression is variable for this gene. Other common genes associated with a later onset LCA include *RPGRIP1*, *RDH12*, and *AIPL1*.

Because of the high phenotypic variability in LCA, retinal specialists incorporate multimodal imaging to guide the diagnosis. Optical coherence tomography (OCT) is helpful, but certain genes exhibit relatively preserved outer retinal structures, including that of *GUCY2D* and *AIPL1* [19,20]. *CEP290*-associated LCA often displays a structurally intact foveal outer nuclear layer until the fourth decade of life [21]. Conversely, patients with *RDH12*-associated LCA develop macular excavation on OCT with associated changes in fundus autofluorescence (FAF) [22]. Similarly, *RPE65*-related retinal dystrophy will show reduced or absent FAF findings [23]. Flash ERG and flash VEP may be helpful in detecting lesser-affected photoreceptors to establish the correct diagnosis [24].

Gene replacement therapies utilizing adeno-associated viral (AAV) transmission are continuously expanding and are of significant interest to the field of ophthalmology. In 2017, the FDA approved Luxturna^®^ for the treatment of *RPE65*-associated LCA [25]. Other therapies targeting genes, including *RPGR* [26] and *CEP290* [27], are currently under investigation. Genetic testing is available via next-generation sequencing (NGS) at the IRD service at the University of Minnesota referral center. There are published studies that examine the underlying genetics of cohorts of patients with the LCA phenotype. These have demonstrated regional variability in the genetic underpinnings of the disease. To date, there is no literature that focuses on a Midwest cohort. This study reports the clinical exam findings, diagnostic imaging markers, and the pathogenic or likely pathogenic variants identified utilizing NGS in our LCA patient cohort.

## 2. Results

### 2.1. Demographic Information and Presenting Symptoms

A total of 34 patients, 32 unrelated pedigrees and 2 related pedigrees, diagnosed with an LCA phenotypic spectrum were evaluated between 1 May 2015 and 5 August 2022. The distribution consisted of 32 patients with non-syndromic LCA and one with syndromic LCA, which was identified to be Joubert syndrome. The demographic information of our LCA patient cohort is listed in Table 1. Most patients noted ocular symptoms before the age of 10. Most did not have a known history of retinal dystrophy. The most common symptoms included 27/31 (87.1%) with nyctalopia, 12/22 (54.5%) with photosensitivity/hemeralopia, 15/21 (71.4%) with color vision impairment measured using the Ishihara test, and 28/29 (96.6%) with observed visual field loss from either the patient or parent. The varying denominators reflect the number of subjects for whom this information was available through retrospective chart review. The age of the initial genetic test is reported in Appendix A.

### 2.2. Visual Acuity Results

A total of 28/32 (87.5%) subjects had visual acuity worse than 20/80 in the right eye and 29/32 (90.6%) in the left eye (Table 1). The other two remaining pairs of eyes did not undergo visual acuity testing through retrospective chart review. There was no statistically significant difference between the right and left eyes.

### 2.3. Multimodal Imaging Findings

The ellipsoid zone (EZ) bandwidth ranges and FAF findings between the left and right eye are reported in Table 1. The EZ measurements were taken at the baseline visit and measured with optical coherence tomography (OCT) of the macula. Figure 1 displays representative scans of different severity from four subjects in our cohort. Advanced photoreceptor loss, representing a central preserved ellipsoid zone of less than 1500 µm, was noted in 12/14 (85.7%) of the subjects. All 19 patients who underwent FAF had findings of either a ring of macular hypo/hyper AF or peripheral hypo-AF in at least one eye. Figure 2 illustrates the grading criteria for defining rings of macular hypo/hyper AF and peripheral hypo-AF along with the representative FAF findings in the same subjects from Figure 1. The associated gene mutation for each FAF image is labeled underneath each picture. All patients had vision worse than 20/80 for these corresponding FAF images.

### 2.4. Electroretinogram and Visual-Evoked Potential Findings

In addition to genetic testing, the use of full-field ERG was incorporated for our LCA patients at the referral center clinic to help confirm the diagnostic suspicion of LCA. In total, 27/34 (79.4%) LCA patients in our cohort had ERG performed as part of their initial evaluation. The patients who underwent ERG testing had confirmatory results that indicated either a reduction, complete absence, or non-measurable amplitude response in dark- and light-adapted conditions in both eyes. In terms of maximal response, all patients had reduced or non-measurable amplitudes. Oscillatory potentials were absent in the same patients. These results are further displayed in Table 1. In our cohort, 3/34 (8.8%) patients had VEP testing performed that was ordered based on the clinical decision making of the initial ophthalmologic evaluation. All three patients had non-discernable amplitudes for flash VEP responses.

### 2.5. Genetic Testing Report

A total of 31/34 (91.2%) patients had genetic testing completed at the time of this study. The two patients who did not undergo genetic testing were adults (>18 years old) with LCA that was clinically diagnosed in childhood. They chose not to undergo genetic testing due advanced disease presentation with no available treatment options. All patients who underwent genetic testing had at least one pathogenic and/or likely pathogenic variant identified. Figure 3 shows the distribution of the causative genes found in our LCA patient cohort. The top causative genes include *CEP290* (26%), *CRB1* (13%), *GUCY2D* (13%), *RPE65* (13%), *RDH12* (9%), and *RPGRIP1* (9%). A detailed summary of the genetic testing and corresponding results is provided in Figure 4. Of those who underwent genetic testing, 23/31 (74.2%) patients had a diagnostic pathogenic/likely pathogenic variant identified (Figure 4a). This percentage reflects the diagnostic yield rate for our LCA patient cohort. Within this subgroup, 22 patients were non-syndromic and one was syndromic. The syndromic patient was determined to have Joubert syndrome. Moreover, 3/31 (9.7%) patients only had VUSs on genetic testing, but based on correlative LCA clinical presentation, we were suspicious of the possible diagnostic value of the VUS (Figure 4b). The final 5/31 (16.1%) patients were found to be non-diagnostic (Figure 4c).

### 2.6. Gene Therapy Qualifications

All three of our patients who had a variant in the *RPE65* gene underwent gene-replacement therapy evaluation for Luxturna^®^. It was found that two of them (siblings) qualified for the gene therapy, and both patients had received Luxturna^®^ treatment at the time of this study. The patient who did not qualify was found to have too-advanced disease presentation based on visual acuity and multi-modal imaging findings and did not meet the inclusion criteria for therapy.

### 2.7. Supplemental Data

Appendix A is attached that includes the diagnostic variants identified for each LCA patient who underwent genetic testing. It is written in accordance with the current Human Genome Variation Society (HGVS) nomenclature, the heterozygosity of variants, and the American College of Medical Genetics (ACMG) variant classifications, as assigned by the performing genetic laboratory. The data in this database were extracted from genetic test reports from various Clinical Laboratory Improvement Amendments (CLIAs) and accredited by the College of American Pathologists (CAP)-certified laboratories. We did not perform specific splice predictions and/or in silico predictions.

## 3. Discussion

The data that we report are taken from one of the few studies in the United States, specifically in the Midwest region, that have reviewed a large cohort of LCA patients. This study demonstrates the high diagnostic yield of NGS in a cohort of patients with the LCA phenotype. The emergence of gene therapy underscores the importance of the early evaluation of patients with LCA, EOSRD, SECORD, and early-onset RP because treatment requires patients to present in earlier stages of the disease with signs of RPE preservation.

The diagnostic yield of NGS testing in our cohort was of ~74%—nearly three out of every four patients with a clinical diagnosis of LCA who underwent NGS had at least one causative gene variant identified. This diagnostic yield is higher than a reported study involving a Japanese population (56%) [28], similar to an Italian population (~80) [29], and lower than an Australian population (~90%) [30] that all utilized targeted NGS. To note, these studies did follow up patients with family genetic testing, and some performed WGS or WES to confirm a genetic diagnosis. We followed up with family testing in instances where there was only VUS or to confirm that two pathogenic variants were in the trans configuration. We did not follow up with WGS or WES in any of our patients, which may have increased the diagnostic yield.

It has been previously noted that *CEP290* and *GUCY2D* are the most predominant genetic causes of LCA [25,30]. *CEP290* was the most prevalent gene in our cohort (26%), followed by *RPE65*, *CRB1*, and *GUCY2D*, which each had a prevalence of 13%. In comparison, a recent study on a German cohort of 105 patients found their top causative genes to be *CEP290* (21%), *RPE65* (14%), *CRB1* (11%), and *RDH12* (8%) [31]. A Brazilian cohort of 152 patients with LCA/EOSRD reported that the top causative genes were *CEP290* (21%), *RPE65* (16%), *CRB1* (14%), and *RPGRIP1* (10%) [32]. In a Chinese cohort of 87 LCA patients, their most common causative genes included *GUCY2D* (16%), *CRB1* (12%), and *RPGRIP1* (8%) [33]. The various global studies and their LCA-associated gene variant prevalence are presented in Figure 5.

The most common pathogenic variant in our cohort was *CEP290* c.2991+1655A>G, which was similar to the Australian study [30] and consistent with its label as the founder variant [15,34]. Our cohort had two *RPE65* c.271C>T p. (Arg91Trp) variants, both of which were homozygous in the two individuals who qualified for Luxturna^®^. The only other repeated variant found in our cohort was *CBR1* c.1949G>A p. (Trp650*).

The single syndromic patient in our cohort was diagnosed both clinically and genetically with Joubert syndrome. This patient had confirmed pathogenic variants in the *CEP290* gene. Five other patients in our cohort had diagnostic pathogenic/likely pathogenic variants in the *CEP290* gene. However, each of these patients only had one clinical examination at our IRD referral center and did not display evidence of systemic involvement. It is possible that, with additional follow up, they may develop systemic symptoms that could point to a syndrome.

Eight patients (25%) in our cohort had the LCA phenotype with non-diagnostic genetic testing. Three of these had VUSs in LCA-associated genes (Figure 4b), including *CEP290* and *RDH12*. Based on the clinical presentation, our team is suspicious that the gene variants are diagnostic; however, we would need to complete further family testing or have the patients return for an updated NGS panel for possible confirmation. Further work is being conducted to reclassify VUSs through rule-based algorithms based on the information gathered from clinical history, segregation studies, phenotype information, inheritance patterns, and information in public databases [35]. Furthermore, other fields of medicine are utilizing gene-specific learning models in the post-genetic testing diagnostic analyses to predict variant pathogenicity [36]. These methods and others may address the issue of VUSs and increase the diagnostic yield of NGS testing in the future.

The remaining five patients (Figure 4c) for whom we were unable to identify a causative gene variant(s) through NGS were considered non-diagnostic in our study. One patient had no clear genetic cause for their LCA diagnosis. One patient had NGS that was carried out in 2007 and had only one *CEP290* variant identified, but this specific test ordered at that time only analyzed for one common *CEP290* variant. This person has not pursued additional genetic testing since that time, so it is unknown whether they have a second *CEP290* variant or variants in a different gene causing their LCA. One patient had their genetic testing performed at an outside institution, and clinical documentation commented on two variants in the *ITF140* gene. However, our team did not include this case in Appendix A, as we were unable to verify the official test report. The final two patients in this group both had a single variant in the *GUCY2D* gene. One patient had testing that analyzed 17 genes associated with LCA in 2011, and the other had a 175-gene IRD panel performed in 2014. Neither patient has pursued additional genetic testing. It is unknown whether they have a second unidentifiable *GUCY2D* variant not captured by the current sequencing technology or may have a separate cause for their LCA. WGS or WES may further provide a diagnostic answer for these patients with non-diagnostic results on the initial NGS sequencing panels [37].

Our team considers genetic testing to be a crucial component of all IRD patient work-up, particularly for patients who arrive at our referral retinal service at a young age. An important aspect of our LCA cohort is the timing at which the first genetic testing was performed after their diagnosis. Because LCA is considered an early-onset form of retinal dystrophy, 31 of our patients first reported ocular symptoms before the age of 10. As a result, 20/31 (64.5%) had their genetic testing before the age of 10 when we promoted genetic testing at the initial clinic visit. The diagnostic yield rate for our LCA patient cohort was noted to be ~20% greater than our RP patient cohort [38].

Three patients in our LCA patient cohort had the *RPE65* gene variant. This allowed those patients to be evaluated for Luxturna^®^, the only current FDA-approved gene therapy. The utilization of the wild-type *RPE65* complementary DNA in the RPE target cells of animal models and humans via an adeno-associated viral vector for supplementation therapy has shown to significantly improve visual function in Phase I–III trials [39,40]. Two of our patients received treatment at the time of this study, and one was excluded due to advanced disease presentation. The patient who was excluded was 59 years old at the time of the genetic testing. At the initial clinical evaluation, his visual acuity, FAF, and EZ bandwidth findings were not within the limits for gene therapy qualification. However, it is important to recognize that ~10% of our LCA patient cohort had the opportunity to be evaluated for gene therapy. The emergence of gene therapies is a large area for emerging treatments for what was previously thought to be an incurable disease [41]. This is important, as there are more gene therapies being developed for LCA, as this is an IRD with an earlier disease presentation.

Our research demonstrates the importance of developing a comprehensive IRD database to include the most updated clinical examination findings and genetic testing results. This can help us identify patients who may qualify for upcoming gene therapy clinical trials [41].

Our study has certain limitations. First, this was a retrospective chart review from a single institution; thus, our results may not be generalizable to other populations. However, this provides a point of comparison for other LCA patient cohorts. Furthermore, although our sample size is large, based on the known prevalence and geolocation of our patient population, more genetic data are needed to understand how gene mutations relate to specific findings on either OCT or FAF imaging. Future studies may address the rate of EZ bandwidth degeneration over time and how this may correlate to the gene mutations that exist in our patient population.

## 4. Methods

### 4.1. Study Design

This is a retrospective study of a cohort of individuals with LCA evaluated at the IRD Clinic at the University of Minnesota (UMN/M Health System). All patients seen between 1 May 2015 (the date our institution implemented its current electronic medical record system) and 5 August 2022 were included, according to our IRB STUDY00012478. All eligible patients were included regardless of age, race, or gender. Patients within our hospital system may opt out of inclusion in retrospective chart reviews at the time of initial consent for service. All patients who opted out were excluded from this analysis.

### 4.2. Defining LCA

A well-cited classification system developed by Foxman distinguishes the difference between complicated/uncomplicated LCA, juvenile RP, and early-onset RP [42]. However, this study does not consider the patient’s genetic profile since it was developed in 1985. The main criteria for this classification system utilize the onset of disease and ERG results. To note, there is ambiguity in this LCA classification system based on the non-reproducible ERG results and the difficulties in determining the exact onset of disease presentation [31,43]. Because of the clinical and genetic heterogeneity, in addition to the lack of a clear definition for LCA, we characterized our cohort as a continuum of phenotypes, considering the personal history, clinical examination, multimodal imaging findings, electroretinogram, and genetic testing results.

### 4.3. Database

The REDCap© software 13.7.29 platform was utilized to curate an LCA database, and a survey was created to input data retrospectively. Clinical information was collected using EPIC©, our institution’s electronic healthcare record system. Data collection included baseline demographic information, ocular history, and pertinent exam findings. Each patient received a randomized numerical assignment, accompanied by their medical identification number. Any question addressed in the survey that was not found in the patient chart was labeled as ‘unknown’.

The entry questions included past ocular history, family history of IRDs, baseline ocular exam, genetic report, and multimodal imaging results. The age at which the patient first noted eye symptoms was recorded (ranges include <10, 10–19, 20–40, and >40 years of age). The self-identified race options listed in our electronic health record included Caucasian, African descent, Latinos, Asian, other, and declined to state. The subjective symptoms included nyctalopia, hemeralopia/photosensitivity, blurry vision, and visual field loss. Visual acuity was recorded categorically as 20/40 or better, worse than 20/40 but better than or equal to 20/80, and worse than 20/80. The presence of visual field loss was documented based on perceived findings from the patient or parent. We did not include formal perimetry because we infrequently order these tests due to the young age at which these patients are often diagnosed.

### 4.4. Multimodal Imaging

The multimodal imaging and diagnostic testing for patients in our cohort included ultra-widefield FAF (Optos^®^, Nikon, Dunfermaline, UK), OCT imaging (Heidelberg-Spectralis^®^, Heidelberg Engineering, Inc, Franklin, MA, USA), ERG E3 system, Diagnosys LLC, Lowell, MA, USA and VEP E3 Diagnosys LLC, Lowell, MA, USA or RETeval^®^ LKC technologies, Gaithersburg, MD, USA. The decision to order an image or test was based on the clinical impression from the retinal specialist (SM). Not all patients received each imaging modality or diagnostic test. The presence of hypo- vs. hyper autofluorescence in the macula on FAF was recorded along with peripheral hypo autofluorescence to identity areas suggestive of oxidative stress and increased metabolic activity [44]. OCT imaging was used to analyze the macular ellipsoid zone (EZ) bandwidth, as this has been a useful marker in monitoring disease progression and structural damage in RP patients [45]. The marking of the EZ endpoint locations was measured manually. Two graders, including one retina specialist, evaluated the EZ bandwidth for all patients. Subjects with an EZ bandwidth of less than 1500 μm were considered to have advanced photoreceptor loss. It should be noted that our OCT imaging allows the detection of the EZ bandwidth up to 6000 μm in length. Measurements of EZ bandwidth preservation past this measurement could not be definitively determined.

### 4.5. Genetic Testing

Our approach for determining whether a patient’s genetic testing results were diagnostic aligns with the ACMG standards and guidelines for the interpretation of sequence variants [46]. Genetic testing was offered to all LCA patients during their initial IRD evaluation. Clinical and family histories were collected, and subsequent genetic counseling was provided for most patients. Whether genetic testing was performed, along with the genetic variant(s) identified, was recorded for each patient. Thirty-one patients had genetic testing performed via a next-generation sequencing (NGS) inherited retinal disease panel or single-gene testing.

The use of NGS has been shown to be an effective method for detecting pathogenic gene variants previously at our institution [38]. NGS panel reports provide a list of genetic variants identified in the patient sample that could be associated with an IRD. The genetic testing laboratories provided in this study are accredited by the College of American Pathologists (CAP), are Clinical Laboratory Improvement Amendments (CLIA)-certified, and utilize the American College of Medical Genetics and Genomics (ACMG) variant classification guidelines to classify each variant identified. Common laboratories used in our cohort included Invitae Laboratory, Blueprint Genetics, PreventionGenetics, and the University of Minnesota Molecular Diagnostic Laboratory. The percentage of patients who utilized each gene panel was considered, and an analysis of the pathogenic/likely pathogenic diagnostic yield rate was calculated for each of the listed gene panels, along with the presence of variants of uncertain significance (VUSs) (Appendix A). The number of genes analyzed in each panel varied depending on the time of genetic testing and the company itself. The range of genes included in genetic panels ranged between 1 and >330 genes. Because of advances in testing capabilities with time, patients evaluated towards the end of the study period may have had more genes tested.

The outcomes of genetic testing include pathogenic/likely pathogenic variants for genes known to cause the LCA phenotype, pathogenic/likely pathogenic variants for genes associated with phenotypes that the patient does not have, VUSs, or no variants. Patients were identified as having diagnostic genetic results if (1) the patient had sufficient pathogenic/likely pathogenic genetic variant(s), (2) the genetic variants were consistent with the patient phenotype, and (3) the genetic variants were consistent with the known inheritance pattern. Because all LCA patients exhibit an autosomal-recessive (AR) inheritance pattern, two pathogenic or likely pathogenic variants were needed to be considered diagnostic. Family studies were conducted to confirm the two variants were in the trans configuration (i.e., one variant on each allele) when possible. In other instances, the variants were confirmed to be in the trans configuration based on the sequencing results, homozygosity for the variant, or presumed to be in the trans configuration based on the clinical phenotype. The patient was classified as having carrier status if a single pathogenic variant was identified, but that gene was associated with AR inheritance. There were situations where testing identified a single pathogenic/likely pathogenic variant in addition to a single variant of uncertain significance (VUS) in the same AR gene. These cases were deemed as clinically suspicious, but not as diagnostic for the purposes of this study. For rarer autosomal-dominant inheritance patterns, only one pathogenic/likely pathogenic variant was required. A patient was considered to have negative genetic test results if no genetic variants were identified or if the only variants identified were VUSs. However, it should be noted that none of our LCA patients fell into this category.

## 5. Conclusions

Our cohort of patients with LCA from our referral center in Minnesota in the Midwest region of the United States consisted of primarily non-syndromic LCA. More than 90% of our cohort presented with visual symptoms in early childhood, but only approximately 60% underwent genetic testing before the age of 10. We found a diagnostic yield of ~74% with NGS testing, which is comparable to other reported rates in other regions of the world. The most common causative genes identified included *CEP290*, *RPE65*, *CRB1*, and *GUCY2D*. Approximately 10% of our cohort had the *RPE65* gene variant that is treatable with FDA-approved therapy.

## Figures and Tables

**Figure 1 ijms-25-01253-f001:**
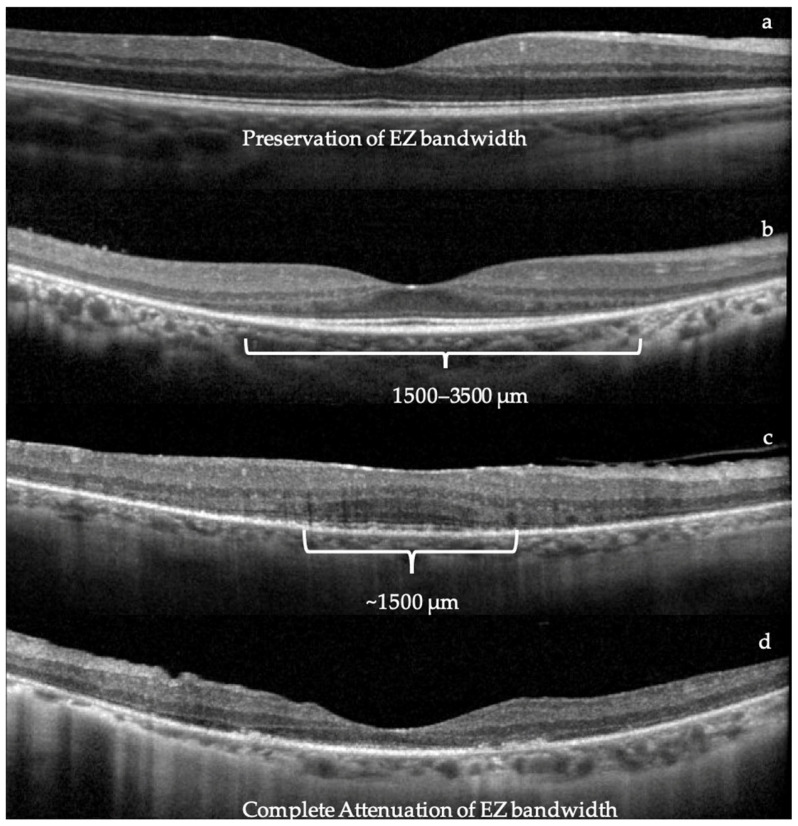
Examples of ellipsoid zone (EZ) bandwidth: (**a**) normal EZ; (**b**–**d**) progression of EZ degeneration to (**b**) EZ 1500–3500 µm, (**c**) EZ 1500 µm, and (**d**) complete attenuation of EZ.

**Figure 2 ijms-25-01253-f002:**
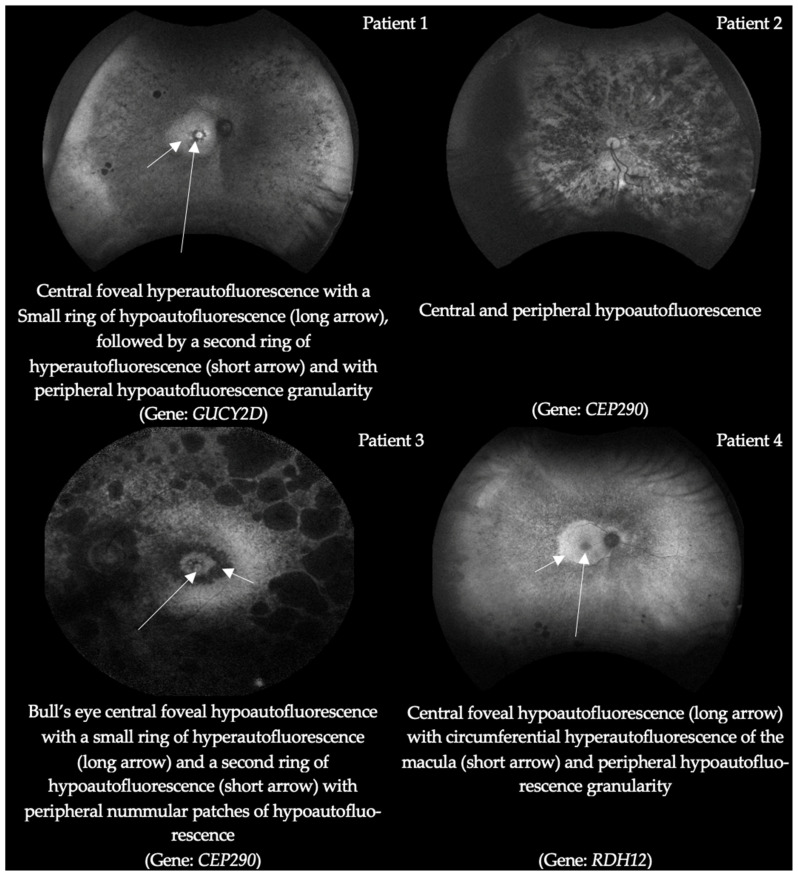
Example grading criteria for FAF imaging.

**Figure 3 ijms-25-01253-f003:**
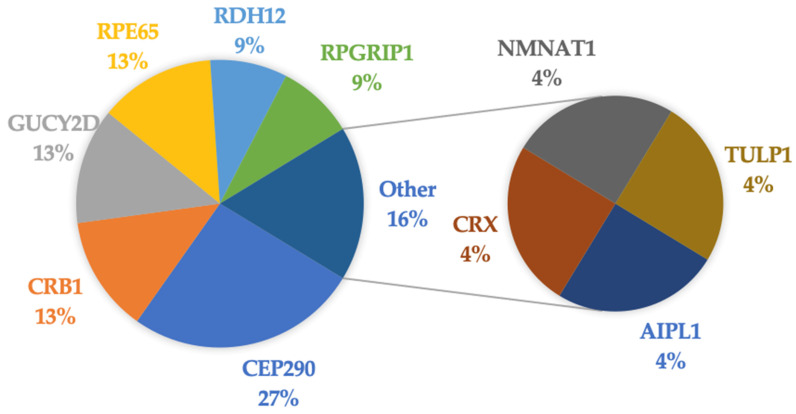
Distribution of causative genes identified in the LCA patient cohort.

**Figure 4 ijms-25-01253-f004:**
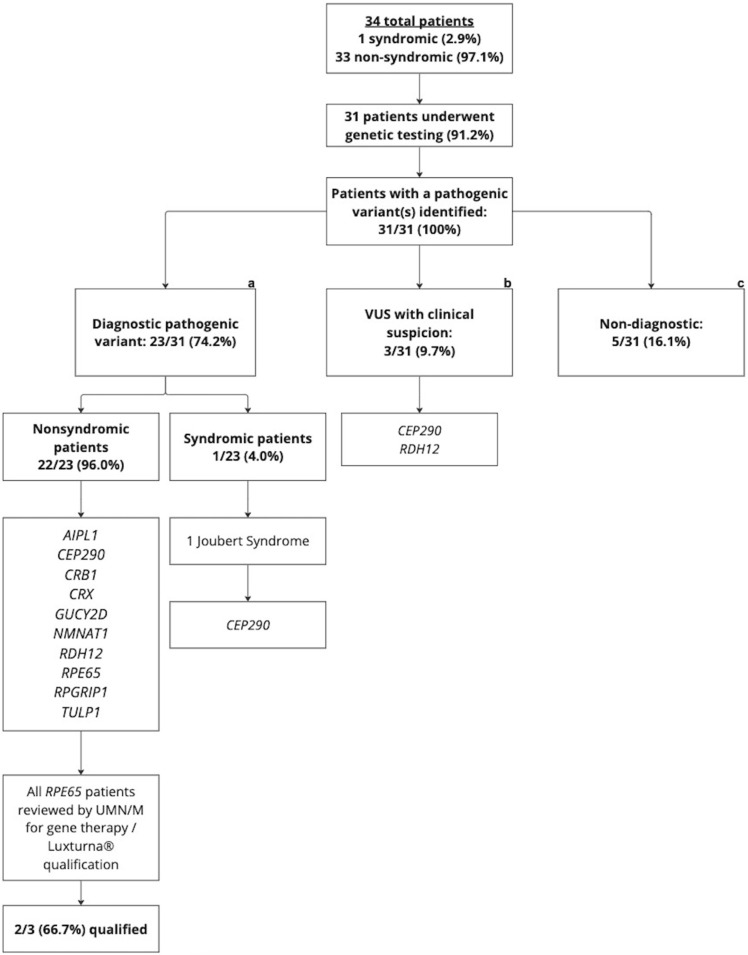
Detailed summary of the LCA patients who underwent genetic testing. (**a**) Nonsyndromic and syndromic patients in our cohort with a diagnostic pathogenic/likely pathogenic variant identified on genetic testing (**b**) Patients with only variable(s) of uncertain significance on genetic testing but suspicious of the possible diagnostic value based on the clinical presentation (**c**) Patients that were found to be non-diagnostic on genetic testing.

**Figure 5 ijms-25-01253-f005:**
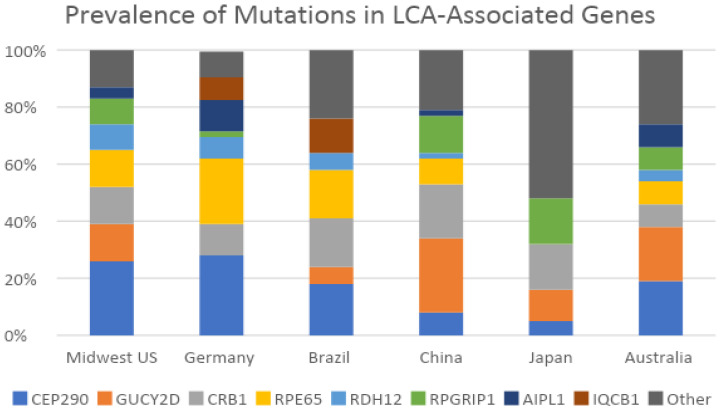
Prevalence of mutations in LCA-associated genes. The studies included the Midwest US (our cohort), Germany [31], Brazil [32], China [33], Japan [28], and Australia [30].

**Table 1 ijms-25-01253-t001:** Patient cohort demographic information.

Sex			Total Cases	
Male			19	
Female			15	
Self-identified Race				
Caucasian			22	
African descent			2	
Latinos			2	
Asian			4	
Declined to state			4	
	Age range of cohort		Age range of ocular symptom onset	Age range of initial genetic testing
Before the age of 10	7 (20.6%)		31 (91.2%)	20 (58.8%)
Between the ages of 10 and 19	16 (47.1%)		1 (2.9%)	4 (11.8%)
Between the ages of 20 and 40	6 (17.6%)		1 (2.9%)	5 (14.7%)
After the age of 40	5 (14.7%)		1 (2.9%)	3 (8.8%)
Visual Acuity				
Snellen (LogMAR)	≥20/40		<20/40–≥20/80	<20/80
	(≤0.3)		(>0.3–≤0.6)	(>0.6)
Right eye	0		4	28
Left eye	0		3	29
EZ width	<1500 μm		1500–3500 μm	3501–6000 μm
Right eye	12		0	2
Left eye	12		0	2
FAF Findings	Normal	Macula ring of hypo-AF	Peripheral hypo-AF	Macula ring of hyper-AF
Right eye	0	7	17	8
Left eye	0	7	17	8
ERG Findings	Dark-adapted conditions	Light-adapted conditions	Maximal response	Oscillatory potential
Right eye	Non-measurable: 25 Reduced: 2	Non-measurable: 26 Reduced: 1	Non-measurable: 24 Reduced: 3	Absent: 25 Reduced: 2
Left eye	Non-measurable: 26 Reduced: 1	Non-measurable: 20 Reduced: 7		

## Data Availability

Data are contained within the article and Appendix A.

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
