# Peer review of "The Clinical Findings, Pathogenic Variants, and Gene Therapy Qualifications Found in a Leber Congenital Amaurosis Phenotypic Spectrum Patient Cohort"

_ijms, 2024, doi:10.3390/ijms25021253_

Round 1

Reviewer 1 Report

Comments and Suggestions for Authors

Please find some comments on the following manuscript:

What are the clinical characteristics and underlying genetic variants present in a Leber congenital amaurosis (LCA) patient cohort evaluated at the inherited retinal disease (IRD) clinic at the University of Minnesota (UMN) / M Health System, and how do these factors contribute to disease presentation and progression in the Midwest region of the United States?

The research is both original and relevant as it explores the clinical characteristics and genetic variants in a Leber congenital amaurosis (LCA) patient cohort from the Midwest region of the United States. The study's emphasis on regional factors, comprehensive diagnostic methods, and treatment outcomes fills a gap in the field by providing unique insights into LCA in a specific geographic context and offering a holistic approach to understanding the disease.

Authors could not able to covey the significance of this innovation and depth of problem. For instance, only 34 patients, 32 unrelated pedigrees and two related pedigrees are taken in this experiment are included, diagnosed between May 1st, 2015 – August 5th, 2022. It seems disease is not frequent and rare. If it is not rare then authors need to collect more data. Induction of more data will help to get necessary and more generalized conclusions.

Captions of the Figures should be provided below that figure. Figure 4 should be divided into 4 sub-parts (a) to (d). Instead of writing (Figure 4, Column b), it should convey like 4(a), 4(b), etc. or authors can think of better methods for good representation.

1.       Authors could not able to covey the significance of this innovation and depth of problem. For instance, only 34 patients, 32 unrelated pedigrees and two related pedigrees are taken in this experiment are included, diagnosed between May 1st, 2015 – August 5th, 2022. It seems disease is not frequent and rare. If it is not rare then authors need to collect more data. Induction of more data will help to get necessary and more generalized conclusions.

2.       Possible future works should be discussed in short based on the limitations of the current study.

3.       The results and discussion are quite weak. For example, it is stated as, “However, our team did not include this case in our gene list, as we do not have a copy of the test report in our electronic health record to verify the identified variants.”  Authors should only focus on what is available. Please check.

4.       Captions of the Figures should be provided below that figure.

5.       Figure 4 should be divided into 4 sub-parts (a) to (d). Instead of writing (Figure 4, Column b), it should convey like 4(a), 4(b), etc. or authors can think of better methods for good representation.

6.       The main negative aspect of this manuscript is the lack of comprehensive experimentation i.e. graphs and tables. Also, there can be good state-of-the-art comparison which will show the significance of the study.

7.       Authors can list the limitations of this study and based on it; future works can be given.

8.       The paper organization should be better. Introduction—Material and methods----experimental results----discussion---conclusion

9.       Redundant references can be removed.

10.    Authors can check for minor grammatical errors and typo mistakes.

11.    Authors should follow the complete guidelines for the preparation of this manuscript.

12.    Keywords should be reduced so that only significant keywords would be highlighted.

Comments on the Quality of English Language

Minor editing of the English language required

Author Response

January 16, 2024

Special Issue: "Retinal Diseases and Macular Degeneration: Cell Biology and Molecular Genetics"

IJMS (International Journal of Molecular Sciences)

ID: ijms-2800734

Dear Editors,

Thank you for your email regarding our manuscript: “The Clinical Findings, Pathogenic Variants, and Gene Therapy Qualifications Found in a Leber Congenital Amaurosis Phenotypic Spectrum Patient Cohort”. We would like to thank the reviewers for their constructive comments. We have revised the manuscript in accordance with their suggestions and have addressed all the concerns. The updated manuscript also included edits that address the comments.

Please find below our responses to the reviewers’ comments. The original comments are followed by our responses. In the revised manuscript, changes are tracked and appear as red/blue text.

Thank you for your time and consideration.

Sincerely,

Sandra R Montezuma                                                             Richard N. Sather III

smontezu@umn.edu                                                               sathe130@umn.edu

Professor of Ophthalmology                                                  Medical Student/Clinical Researcher

Knobloch Endowed Chair

Co-authors: Jacie Ihinger, MS (Jacqueline.Ihinger@fairview.org) ; Michael Simmons, MD (simmo720@umn.edu) ; Glenn Lobo, PhD (lobo0023@umn.edu)

University of Minnesota, Twin Cities, MN, USA

Department of Ophthalmology and Visual Neuroscience 

516 Delaware St SE, Minneapolis, MN 55455

Reviewer 1:

Comment 1: Authors could not be able to covey the significance of this innovation and depth of problem. For instance, only 34 patients, 32 unrelated pedigrees and two related pedigrees are taken in this experiment are included, diagnosed between May 1st, 2015 – August 5th, 2022. It seems disease is not frequent and rare. If it is not rare then authors need to collect more data. Induction of more data will help to get necessary and more generalized conclusions.

Response 1: As mentioned in the second sentence of the introduction, the prevalence of LCA is approximately 2-3 cases per 100,000 births. This makes this disease very rare compared to that of other retinal dystrophies (i.e. retinitis pigmentosa, Stickler, Stargardt). Overall, our patient cohort is large based on only an analysis of the cases from the Midwest region of the United States. The date range we included was based on our IRB approval. These are the patients that we could find in our medical record system based on the set date range.

Comment 2: Possible future works should be discussed in short based on the limitations of the current study.

Response 2: Possible future works have been added in short to the last paragraph of the discussion section. Thank you.

Comment 3: The results and discussion are quite weak. For example, it is stated as, “However our team did not include this case in our gene list, as we do not have a copy of the test report in our electronic health record to verify the identified variants.”  Authors should only focus on what is available. Please check.

Response 3: Thank you for this comment. Our team was trying to convey that, although the patient had a reported known diagnostic variant from an outside institution, we did not include them under Figure 4a but rather Figure 4c because we did not have a copy of their genetic report. We included this comment if the reader was trying to compare Figure 4 to Supplemental Table 1. The wording of that sentence has been modified. The remainder of the paper has been checked and focuses on only information that is available.

Comment 4: Captions of the Figures should be provided below that figure.

Response 4: The captions of the figures are now listed below each figure.

Comment 5: Figure 4 should be divided into 4 sub-parts (a) to (d). Instead of writing (Figure 4, Column b), it should convey like 4(a), 4(b), etc. or authors can think of better methods for good representation.

Response 5: Thank you for the suggestion. We decided to keep the layout of Figure 4 the same, but changed the naming convention to Figure 4a, Figure 4b, and Figure 4c throughout the manuscript. 

Comment 6: The main negative aspect of this manuscript is the lack of comprehensive experimentation i.e. graphs and tables. Also, there can be good state-of-the-art comparison which will show the significance of the study.

Response 6: Thank you for the overall feedback. The purpose of our study was to demonstrate how an in-depth retrospective clinical review can promote an understanding of the pathogenic variants, clinical characteristics, and imaging results for a LCA patient population. Our paper is unique in that it is the first study in the Midwest region to examine this specific patient population. It has good clinical applicability to ophthalmologists who care for patients with inherited retinal diseases. Our goal is to help clinicians understand the benefits of utilizing a comprehensive database to communicate the topics discussed in our paper. As a result, this goes away from an experimental basic science approach and looks more at the application of genetic testing from a clinical standpoint. We do, however, consider the current literature that exists on this topic and subsequently do a formal comparison of our patient cohort to other countries for the same condition (Figure 5).

Comment 7: Authors can list the limitations of this study and based on it; future works can be given.

Response 7: Thank you for this suggestion. The limitations of this study and future studies for our team were provided in the final paragraph of the discussion section.

Comment 8: The paper organization should be better. Introduction—Material and methods----experimental results----discussion---conclusion

Response 8: We initially formatted our paper in this order, but this journal requires this specific format for publication.

Comment 9: Redundant references can be removed.

Response 9: Redundant references have been removed in our paper.

Comment 10: Authors can check for minor grammatical errors and typo mistakes.

Response 10: Minor grammatical errors and type mistakes have been adjusted in our paper.

Comment 11: Authors should follow the complete guidelines for the preparation of this manuscript.

Response 11: The complete guideline has been reviewed. Our assistant editor, Bab Wang, made some formatting arrangements for our manuscript to meet journal standards of publication.

Comment 12: Keywords should be reduced so that only significant keywords would be highlighted.

Response 12: The keywords have been reduced to only significant keywords.

Reviewer 2 Report

Comments and Suggestions for Authors

The retrospective chart review study reports the clinical and genetic characteristics of LCA patients, seen in an academic center/tertiary IRD clinic in Midwest US between 2015-2022. Overall, the study is very well presented with adequate introduction and relevant discussion. Suggested edits below will improve the overall impact and clarity of the manuscript.

Line 73: "These variants produce" - Statement is ambiguous. Can specify RPE65.

Line 88: OCT is Optical Coherence Tomography NOT "Ocular"

Line 99/100: Investigational therapies - Need Citation/REFs

Line 121: Table 1 - Please include ERG/VEP findings

Line 137: How are these OCT images related to FAF images in Fig. 2? (Ideally OCT and FAF shown are from same patients, so that interpreting it is easy for the readers). If they are from same patients label them as P1-P4 and label the same in Fig. 2 also.

Line 172: Consider including the gene and the VA in this fundus photo. Could use arrow and arrow heads to point to Hyper and Hypo AF.

Line 208: Typo "Luxterna"

Figure 4: Column A should either read 'reviewed by UMN/M' or 'reviewed for gene therapy/Luxturna'.

Line 225/Discussion section: Consider including the significance of findings of the imaging (OCT/FAF), specifically with regards to LCA phenotypes. For example, the most common CEP290/GUCY2D mutations - do they have characteristic OCT/FAF features?

Line 319: Typo: Should read 'Evaluated for'

Author Response

January 16, 2024

Special Issue: "Retinal Diseases and Macular Degeneration: Cell Biology and Molecular Genetics"

IJMS (International Journal of Molecular Sciences)

ID: ijms-2800734

Dear Editors,

Thank you for your email regarding our manuscript: “The Clinical Findings, Pathogenic Variants, and Gene Therapy Qualifications Found in a Leber Congenital Amaurosis Phenotypic Spectrum Patient Cohort”. We would like to thank the reviewers for their constructive comments. We have revised the manuscript in accordance with their suggestions and have addressed all the concerns. The updated manuscript also included edits that address the comments.

Please find below our responses to the reviewers’ comments. The original comments are followed by our responses. In the revised manuscript, changes are tracked and appear as red/blue text.

Thank you for your time and consideration.

Sincerely,

Sandra R Montezuma                                                             Richard N. Sather III

smontezu@umn.edu                                                               sathe130@umn.edu

Professor of Ophthalmology                                                  Medical Student/Clinical Researcher

Knobloch Endowed Chair

Co-authors: Jacie Ihinger, MS (Jacqueline.Ihinger@fairview.org) ; Michael Simmons, MD (simmo720@umn.edu) ; Glenn Lobo, PhD (lobo0023@umn.edu)

University of Minnesota, Twin Cities, MN, USA

Department of Ophthalmology and Visual Neuroscience 

516 Delaware St SE, Minneapolis, MN 55455

Reviewer 2:

Comment 1: Line 73: "These variants produce" - Statement is ambiguous. Can specify RPE65.

Response 1: The wording of this sentence has been changed. Thank you.

Comment 2: Line 88: OCT is Optical Coherence Tomography NOT "Ocular"

Response 2: Thank you for this observation. The error has been fixed.

Comment 3: Line 99/100: Investigational therapies - Need Citation/REFs

Response 3: Citations have been added to address the investigational therapies.

Comment 4: Line 121: Table 1 - Please include ERG/VEP findings

Response 4: Thank you for this suggestion. The ERG results have been added to Table 1. We did not include the VEP results because only three patients had this testing done, all of which showed non-discernable amplitudes for flash VEPs. This comment was included in the results section accordingly.

Comment 5: Line 137: How are these OCT images related to FAF images in Fig. 2? (Ideally OCT and FAF shown are from same patients, so that interpreting it is easy for the readers). If they are from same patients label them as P1-P4 and label the same in Fig. 2 also.

Response 5: Thank you for this suggestion. Patients 1 – 4 have been labeled between Figure 1 and Figure 2. There was a statement made about this in the results section to help with easier interpretation for the readers.

Comment 6: Line 172: Consider including the gene and the VA in this fundus photo. Could use arrow and arrow heads to point to Hyper and Hypo AF.

Response 6: The associated gene mutation has been added for each fundus photo. The arrows have been added to help point to the areas of hyper – and hypo – AF. A comment about the visual acuity was added to the results section.

Comment 7: Line 208: Typo "Luxterna"

Response 7: Thank you for this observation. The edit has been made.

Comment 8: Figure 4: Column A should either read 'reviewed by UMN/M' or 'reviewed for gene therapy/Luxturna'.

Response 8: This suggestion has been noted and changed appropriately in Figure 4.

Comment 9: Line 225/Discussion section: Consider including the significance of findings of the imaging (OCT/FAF), specifically with regards to LCA phenotypes. For example, the most common CEP290/GUCY2D mutations - do they have characteristic OCT/FAF features?

Response 9: Thank you for this comment. At this moment, it is difficult to find a characteristic OCT/FAF that is unique specific gene mutations. When the sample size expands for a particular mutation, we will consider doing a future study to identify particular features on imaging. We included a statement on this in the discussion section at the end.

Comment 10: Line 319: Typo: Should read 'Evaluated for'

Response 10: Thank you for this observation. The edit has been made.

Round 2

Reviewer 1 Report

Comments and Suggestions for Authors

I have thoroughly reviewed the manuscript and author's responses, and I am satisfied with the revisions made. The manuscript is well-written, clear, and relevant to the research question with significant results. Based on my assessment, I recommend it for publication.